Ultraconserved elements (UCEs) illuminate the population genomics of a recent, high-latitude avian speciation event

Winker Kevin 1 kevin.winker@alaska.edu
http://orcid.org/0000-0001-7725-3637 Glenn Travis C. 2
http://orcid.org/0000-0002-1943-0217 Faircloth Brant C. 3
1 University of Alaska Museum & Department of Biology and Wildlife, University of Alaska Fairbanks , Fairbanks, AK , USA
2 Department of Environmental Health Science and Institute of Bioinformatics, University of Georgia , Athens, GA , USA
3 Department of Biological Sciences and Museum of Natural Science, Louisiana State University , Baton Rouge, LA , USA
Wink Michael
Electronic publication date: 2018 Oct 5
Publication date: 2018
Volume: 6
Electronic Location ID: e5735
Received 2018 Apr 4; Accepted 2018 Sep 5
Copyright: © 2018 Winker et al.
Copyright year: 2018
Copyright holder: Winker et al.
License: This is an open access article distributed under the terms of the Creative Commons Attribution License, which permits unrestricted use, distribution, reproduction and adaptation in any medium and for any purpose provided that it is properly attributed. For attribution, the original author(s), title, publication source (PeerJ) and either DOI or URL of the article must be cited.
License URL: https://creativecommons.org/licenses/by/4.0/

Keywords: Conserved loci, Genome sampling, Speciation, Passeriformes, Plectrophenax

Funding: National Science Foundation DEB-1242267-1242241-1242260 National Center for Genome Analysis Support (Trinity on Galaxy for assembly) and the Research Computing Systems of the University of Alaska Fairbanks (for analyses) provided computational support The National Science Foundation (DEB-1242267-1242241-1242260) supported this study. The National Center for Genome Analysis Support (Trinity on Galaxy for assembly) and the Research Computing Systems of the University of Alaska Fairbanks (for analyses) provided computational support. The funders had no role in study design, data collection and analysis, decision to publish, or preparation of the manuscript.

==============================
Using a large, consistent set of loci shared by descent (orthologous) to study relationships among taxa would revolutionize among-lineage comparisons of divergence and speciation processes. Ultraconserved elements (UCEs), highly conserved regions of the genome, offer such genomic markers. The utility of UCEs for deep phylogenetics is clearly established and there are mature analytical frameworks available, but fewer studies apply UCEs to recent evolutionary events, creating a need for additional example datasets and analytical approaches. We used UCEs to study population genomics in snow and McKay’s buntings (Plectrophenax nivalis and P. hyperboreus). Prior work suggested divergence of these sister species during the last glacial maximum (∼18–74 Kya). With a sequencing depth of ∼30× from four individuals of each species, we used a series of analysis tools to genotype both alleles, obtaining a complete dataset of 2,635 variable loci (∼3.6 single nucleotide polymorphisms/locus) and 796 invariable loci. We found no fixed allelic differences between the lineages, and few loci had large allele frequency differences. Nevertheless, individuals were 100% diagnosable to species, and the two taxa were different genetically (FST = 0.034; P = 0.03). The demographic model best fitting the data was one of divergence with gene flow. Estimates of demographic parameters differed from published mtDNA research, with UCE data suggesting lower effective population sizes (∼92,500–240,500 individuals), a deeper divergence time (∼241,000 years), and lower gene flow (2.8–5.2 individuals per generation). Our methods provide a framework for future population studies using UCEs, and our results provide additional evidence that UCEs are useful for answering questions at shallow evolutionary depths.

Introduction

Among non-model organisms, population genetic studies have used a diverse set of markers, tending to concentrate on those with sufficiently high substitution rates to provide useful data at shallow levels of evolutionary divergence, for example, from the populations-to-species levels (Avise, 1994; Hillis, Moritz & Mable, 1996; Pearse & Crandall, 2004). This approach usually provides answers to the specific questions asked by researchers, but the historic focus on markers with high substitution rates has produced studies that include relatively few loci and often have little to no overlap with loci used for other taxa. This lack of consistency in the loci used across different studies compromises our ability to make direct comparisons of population genetic parameters among taxa (e.g., in divergence statistics and in estimates of gene flow and effective population sizes). Improvements in sequencing platforms and genomic data collection approaches are changing this general pattern by enabling us to efficiently collect much larger samples of the genome, up to and including whole-genome sequences (Ellegren, 2014). However, the sheer quantity of data obtained from whole-genome sequencing can require excessively long computation times, and may be overkill for many questions. The parallel difficulties of collecting a moderate sample of the genome from identical loci across diverse species argue for a sequence data collection approach that (a) subsamples the genome to (b) obtain orthologous markers across a broad taxonomic scope. This type of approach would provide a tractable number of loci for analyses while improving among-study comparisons and larger-scale comparative meta-analyses. Ultraconserved elements (UCEs) are one class of genome-wide marker that might provide a solution to these problems.

Ultraconserved elements are conserved sequences shared among divergent animal genomes (Bejerano et al., 2004; Siepel et al., 2005; Stephen et al., 2008; Janes et al., 2011), and many UCE loci are likely to be involved in controlling gene expression (Marcovitz, Jia & Bejerano, 2016). UCEs in vertebrates show little overlap with most types of paralogous genes, and, as a marker class, UCE loci are broadly distributed across the genome and are typically transposon-free (Derti et al., 2006; Simons et al., 2006; McCormack et al., 2011; Harvey et al., 2016). We focus on the set of UCEs previously defined for tetrapods and now in widespread use (McCormack et al., 2011; Faircloth et al., 2012). Outside of their functional relevance, UCE loci have demonstrated utility for recovering deeper-level phylogenetic relationships (McCormack et al., 2013; Faircloth et al., 2015; Gilbert et al., 2015) and shallower-level genus and population relationships (Smith et al., 2014; Harvey & Brumfield, 2015; Leaché et al., 2015; Harvey et al., 2016; Manthey et al., 2016; Oswald et al., 2016; Mason et al., 2018). Although UCEs are highly conserved at their core, which enables universal capture of loci across diverse groups of organisms (Faircloth et al., 2012, 2015; Faircloth, 2013; Starret et al., 2017), lower levels of purifying selection away from the core allow substitutions to accumulate in the flanking regions. Using human genome data, Faircloth et al. (2012) demonstrated that the increased variation in UCE flanking sequence might be adequate to make these loci useful for questions at shallow levels of divergence, and this hypothesis has been supported by subsequent empirical studies (Smith et al., 2014; Harvey & Brumfield, 2015; Harvey et al., 2016; Oswald et al., 2016; Mason et al., 2018). However, the utility of UCE loci for studying population genetics, population divergence, and/or incipient speciation is only beginning to be tested, and both the value and challenges of using UCEs at these shallow levels remain underexplored.

Here, we examine the utility of UCEs for studying the population genomics of divergence between two bird species, McKay’s bunting (Plectrophenax hyperboreus) and snow bunting (P. nivalis). McKay’s buntings breed on remote islands in the Bering Sea (where our samples are from) and are the highest-latitude endemic songbirds; their range is restricted to the North Pacific region. Snow buntings breed throughout the rest of the high-latitude Holarctic (our samples are from the southern edge of the Being sea on the Alaska Peninsula and an Aleutian island; Table S1). McKay’s bunting is thought to have arisen ∼18–74 Kya during the last glacial maximum through divergence from snow buntings, and previous work suggests gene flow between the two may be ongoing (Maley & Winker, 2010). These species are interesting to study using UCEs because prior work (Maley & Winker, 2010) enables us to compare population genetic statistics derived from UCEs versus traditional population genetic markers (mtDNA sequence and amplified fragment length polymorphisms, AFLPs). These species also allow us to test the utility of UCEs for studying very shallow divergences between sister lineages where gene flow may be ongoing.

Methods

Laboratory

We extracted DNA from muscle tissue of eight specimens (four of each species) studied by Maley & Winker (2010) using proteinase K digestion (100 mM Tris pH 8, 50 mM EDTA, 0.5% SDS, one mg/mL proteinase K) followed by SPRI bead purification (Rohland & Reich, 2012; Table S1). We chose this sample size (and our sequencing depth) to ensure that we could confidently call both alleles for each individual in each population to achieve eight sequences per population at each locus, which Felsenstein (2005) considered to be the optimum sample size for coalescent-based analyses. Following DNA extraction, we prepared dual-indexed DNA libraries for each sample using methods described in Glenn et al. (2017). After library preparation, we quantified each library using a Qubit fluorimeter (Invitrogen, Inc., Carlsbad, CA, USA), and we combined eight libraries into equimolar pools of 500 ng each (62.5 ng/library). We enriched the pool of eight samples for 5,060 UCE loci using the Tetrapods-UCE-5Kv1 kit from MYcroarray following version 1.5 of the UCE enrichment protocol and version 2.4 of the post-enrichment amplification protocol (http://ultraconserved.org/) with HiFi HotStart polymerase (Kapa Biosystems, Wilmington, MA, USA) and 14 cycles of post-enrichment PCR. We then quantified the fragment size distribution of the enriched pool on a Bioanalyzer (Agilent, Inc., Santa Clara, CA, USA) and qPCR quantified the enriched pool using a commercial kit (Kapa Biosystems, Wilmington, MA, USA). We combined the enriched pool of eight bunting samples with enriched pools from other birds at equimolar ratios, and we sequenced the resulting pool using one lane of paired-end 150 bp (PE150) sequencing on an Illumina HiSeq 2500 (Illumina, Inc., San Diego, CA, USA; UCLA Neuroscience Genomics Core).

Bioinformatics

After sequencing, we demultiplexed the sequencing reads using bcl2fastq version 1.8.4 (Illumina, Inc., San Diego, CA, USA), and we trimmed the demultiplexed reads for adapter contamination and low-quality bases using a parallel wrapper (Faircloth, 2013) around Trimmomatic (ver. 0.32; Bolger, Lohse & Usadel, 2014). We then combined singleton reads that lost their mate with read 1 files, combined all individual read 1 files (plus singletons) together and all individual read 2 files together, and assembled these two read 1 and read 2 files de novo using Trinity (ver. 2.0.6; Grabherr et al., 2011) on Galaxy (Afgan et al., 2016). After assembling this composite of data from all individuals, we used Phyluce (ver. 1.4.0; Faircloth, 2016) to identify FASTA sequences from orthologous UCEs and remove FASTA sequences from non-UCE loci or potential paralogs. We called the resulting file our reference set of UCE loci, which we used as the reference sequence for calling individual variants.

Next, we used Phyluce and its program dependencies (BWA 0.7.7, Li & Durbin, 2009; SAMtools 0.1.19, Li et al., 2009; Picard 1.106, http://broadinstitute.github.io/picard) to align unassembled, raw reads from individual buntings to the reference set of UCE loci. Specifically, this workflow aligned raw reads on a sample-by-sample basis against the composite reference using the bwa-mem algorithm (preferred for reads >70 bp; Li, 2013); added header information to identify alignments from individual samples; cleaned, validated, and marked duplicates in the resulting Binary Alignment/Map (BAM) file using Picard; and merged all individuals into a single BAM file using Picard. Following preparation of the merged BAM, we used GATK (ver. 3.4-0; McKenna et al., 2010) to identify and realign indels, call and annotate single nucleotide polymorphisms (SNPs) and indels, and mask SNP calls around indels using a GATK workflow described as part of a population genomics pipeline for UCEs developed by Faircloth and Michael Harvey (https://github.com/mgharvey/seqcap_pop). This included restricting data to high-quality SNPs (Q30) and read-back phasing in GATK. After calling and annotating SNPs, we deviated from this workflow by using VCFtools (ver. 0.1.12b; Danecek et al., 2011) to filter the resulting variant call format (VCF) file with the --max-missing (1.0) and --minGQ (10.0) parameters, which created a complete data matrix with a minimum genotype quality (GQ) of 10. We validated that GQ10 data were present for all individuals at all loci by visually assessing alignment data at 17 SNPs among 10 loci using Tablet (ver. 1.15.09.01; Milne et al., 2013). We used GATK’s EMIT_ALL_CONFIDENT_SITES function to ensure that we only retained invariant loci with high quality (rather than missing) data. We then removed variable and invariable loci with incomplete data from downstream analyses, retaining only loci with complete data. This finalized our complete VCF file.

Data analysis

We calculated coverage depths, SNP positions within loci, and SNP-specific and locus-specific FST values on the complete VCF file using VCFtools (ver. 0.1.12b; Danecek et al., 2011). After thinning the VCF file to 1 SNP/locus (which is required in demographic analyses when unlinked variation is important) and converting the VCF file to STRUCTURE format using PGDSpider (ver. 2.1.0.3; Lischer & Excoffier, 2012), we performed tests of Hardy–Weinberg equilibrium and computed observed and expected heterozygosities, homogeneity of variance, population structure (population FST, including a 10,000-replicate G-test; see Goudet et al. (1996)), and the probabilities of each individual’s assignment to a particular population using discriminant analysis of principal components (DAPC) in adegenet (ver. 2.0.1; Jombart & Ahmed, 2011). To calculate nucleotide diversity, average distance between taxa (dxy), and net average distance (dA), we created a concatenated FASTA file of all individual sequences using catfasta2phyml by Johan Nylander (https://github.com/nylander/catfasta2phyml), and we analyzed this file in MEGA (ver. 6; Tamura et al., 2013) using the maximum composite likelihood method.

We estimated recombination using the four-gametes test as implemented in IMgc (Woerner, Cox & Hammer, 2007), which also produces sequence datasets from which the effects of recombination have been removed. Resulting sequences were used for IMa2p (ver. 1.0; Sethuraman & Hey, 2015) analyses to attempt to estimate demographic parameters, but those analyses did not converge under a variety of full- and sub-sampling schemes and are not reported. We nevertheless include the results of IMgc because accounting for recombination is a critical part of workflows using full sequences (i.e., not just SNPs), and these results provide needed insight into the levels of recombination found in UCE loci for studies of this type.

We used Diffusion Approximations for Demographic Inference (δaδi; ver. 1.7.0; Gutenkunst et al., 2009) to infer demographic parameters from the data under a variety of divergence scenarios (models) after excluding Z-linked loci (for δaδi analyses only). Z-linked loci in birds are on the sex chromosome, have a different inheritance scalar from autosomal loci, and sample population sex ratios affect allele frequency estimates (Jorde et al., 2000; Garrigan et al., 2007). We identified Z-linked loci in our data using BLASTn (ver. 2.3.1; Zhang et al., 2000), by aligning the reference set of UCE loci against the zebra finch (Taeniopygia guttata) genome (NCBI Annotation Release 103). We excluded UCE loci that strongly matched (E-values ∼0.0) the zebra finch Z chromosome. After removing Z-linked loci from our complete VCF file, we converted this reduced dataset to biallelic format (which dropped one locus with >2 alleles at a SNP site) and thinned the data to one SNP per locus using VCFtools. Then we converted the resulting VCF file to the joint site frequency spectrum (SFS) format required by δaδi using a PERL script by Kun Wang (https://groups.google.com/forum/#!msg/dadi-user/p1WvTKRI9_0/1yQtcKqamPcJ). Because we lacked an outgroup, we used a folded SFS in our analyses (Gutenkunst et al., 2009), which lacks polarization of SNPs (Fig. S1).

Because we had prior evidence that these species represent two genetic populations (based on taxonomy and results from Maley & Winker, 2010), we used δaδi to infer what general two-population divergence model best fit the data. We then used that model to estimate demographic parameters (i.e., effective population sizes, split time, and migration). We ran six different models spanning the standard possible demographic histories of two populations, five basic and one derivative: (1) neutral (no divergence, or still strongly mixing), (2) split with migration, (3) split with no migration, (4) isolation with bidirectional migration and population growth, (5) isolation with population growth and no migration, and (6) a custom split-bidirectional-migration model (a simple derivative of split-migration; Fig. 1). The neutral, split-with-migration, and isolation-with-migration-and-population-growth models are provided in the δaδi file Demographics2D.py as snm, split_mig, and IM, respectively. The no-migration models (3 and 5 above) use the split_mig and IM models, respectively, with migration parameters set to zero. The split-bidirectional-migration model (figshare https://doi.org/10.6084/m9.figshare.6453125.v1) adds bidirectional migration to the split-migration model to examine potential asymmetry in gene flow.

Figure 1 Population divergence models tested using δaδi.

Population divergence models tested using δaδi, varying from (1) neutral (no divergence), to a series of different two-population models with an ancestral population diverging into two populations (at time T): (2) split with migration (gene flow), (3) split with no migration, (4) isolation with bidirectional migration and population growth, (5) isolation with population growth and no migration, and (6) a derivative of model 2 with bidirectional migration.

We performed a series of optimization runs (8–47 each) of each basic model, adjusting parameters (grid points, upper and lower bounds) to identify high log composite likelihoods. We then ran each model repeatedly, varying parameters within bounds that yielded the highest likelihood during optimization, until three runs yielded the highest observed likelihood value. Our reasoning was that this level of repeatability indicated a best-fit neighborhood for each model. We report this highest observed likelihood, except for poorer models, which yielded variable likelihoods, in which case we averaged and report the highest five values. After identifying the best-fit model based on likelihood values over successive runs, we ran the best-fit model 10 times each with jackknifed datasets to estimate the 95% confidence interval (CI) for each parameter.

We estimated the average per-site substitution rate by BLASTing the FASTA file containing all confidently scored loci (those meeting our quality filters as described in the Bioinformatics section, above) for all individuals (3,431 loci) against the budgerigar (Melopsittacus undulatus) genome (NCBI release 102) and the rifleman (Acanthisitta chloris) genome (NCBI release 100), using time to most recent common ancestor (TMRCA) date estimates of 60.5 Ma (budgerigar) and 53 Ma (rifleman) (Claramunt & Cracraft, 2015). These taxa were chosen as the nearest relatives with complete genomes and fossil-dated nodes available. We imported BLAST results (as a hit table, csv) into a spreadsheet, removed duplicate lower-affinity hits, then summed total length of base pairs, total substitutions, and calculated substitutions per site. This value (substitutions per site) was annualized by multiplying it by 2 TMRCA (e.g., 121 Ma for the total time along the branches of divergence between buntings and budgerigars). To account for the uncertainty associated with using divergence time estimates from distant relatives, we averaged the resulting substitutions per site per year rates (6.83 × 10−10 and 6.67 × 10−10, respectively), and we used the average rate (6.75 × 10−10) to convert parameter estimates obtained from δaδi analyses into biologically relevant estimates of effective population size(s) and split time(s). We converted substitution rates to substitutions/site/generation using a generation time of 2.7 years for snow buntings. We estimated generation time using survival and breeding data from Smith (1994) and the method of Sæther et al. (2005), in which generation time (G) is calculated as G = α + (s/(1 − s)), where α is age of first breeding (1 year) and s is annual adult survival. Finally, because our δaδi analyses used filtered SNPs, our demographic estimates used an adjusted surveyed sequence length of: (total sites surveyed, including invariant sites) × (proportion of SNPs used after thinning to 1 SNP/locus) = 1,103,715 bp.

Results

Assembly produced 632,401 contigs (min = 224 bp, max = 17,453 bp) with a mean length of 396.6 bp (±0.27 bp 95% CI) for a total of 250,802,355 bp. Fully 9,194 contigs were over one Kb in length. After identifying UCE loci and removing potential paralogs, we recovered 4,018 UCE loci. After filtering UCE loci for quality, calling SNPs, phasing (reconstructing haplotypes), and applying additional quality filters, we identified 2,635 loci that contained data for all individuals and were variable. This complete matrix of variable loci included a total of 9,449 SNPs (averaging 3.6 sites per locus). Per-site sequencing depth for these SNPs averaged 26.3 reads (±16.9 SD). An additional 587 loci exhibited variation but the data were not of sufficient quality (i.e., GQ < 10) among all individuals to confidently call both alleles. There were 796 high-quality invariant loci (loci with invariant data, rather than an absence of data), providing a full dataset of 3,431 loci with mean length of 1153.6 bp (±4.95 bp 95% CI). The shortest locus was 228 bp, the longest 2,543 bp, and 2,482 loci were longer than one Kb (Fig. S2). The total length of these loci was 3,957,876 bp. The distribution of SNP variation among loci confidently called for all individuals is given in Fig. 2. Nucleotide diversity (π) was 0.000519 overall, 0.000523 for snow buntings, and 0.000493 for McKay’s buntings.

Figure 2 Distribution of single nucleotide polymorphisms (SNPs) per locus.

Distribution of single nucleotide polymorphisms (SNPs) per locus among 3,431 confidently called loci.

No alleles showed fixed differences (FST = 1.0) between the two populations, and few alleles showed strong segregation. No variable sites had an FST value above 0.9, and there were only three each at 0.86 and 0.72 (Fig. S3; two of these sites were on the same locus). One of the five loci with the highest FST values was Z-linked; all of the others were on different chromosomes (figshare https://doi.org/10.6084/m9.figshare.6453125.v1). There were 128 Z-linked loci among the 2,635 variable loci. As noted, only one showed high FST between the two species. The two populations had an overall FST = 0.034, which was significant (P = 0.03). The average distance between taxa (dxy) was 5.3 × 10−4, and the net average distance (dA) was 2.0 × 10−5. DAPC in adegenet assigned all individuals to their correct taxon of origin (retaining the first four PCs), with 100% probabilities for each, indicating a high level of genomic diagnosability (Fig. S4).

Fully 2,510 loci were in Hardy–Weinberg equilibrium; 124 were not (one was triallelic). McKay’s buntings had fewer unique alleles (4,238) than snow buntings (4,389), concordant with the smaller population size of McKay’s buntings. Bartlett’s test rejected homogeneity of variance between observed heterozygosity (Ho = 0.18, 0.19) and expected heterozygosity (He = 0.20, 0.22), but Ho did not differ from He (t = −3.1653, df = 2,633, P = 1.0).

The four-gametes test suggested that recombination occurred in hundreds of loci. For 405 loci, locus lengths were shortened by IMgc to meet the four-gametes test, and for 252 loci one or more individuals were removed to meet the same criteria (a few of these loci had both done; IMgc automatically performs one or the other or both operations to obtain non-recombinant sequence data). There were thus 15.4–24.9% of variable loci exhibiting patterns indicative of recombination. As noted in the Methods, these sequence data, together with all other unchanged sequences, were not used further; we used only SNP data for further analyses.

In testing our six, two-population models with δaδi, the highest maximum log composite likelihood values were obtained for the split-with-migration model (−112.76), which made it the best-fitting model for these data (model 2 in Fig. 1). We obtained successively lower likelihood values for the neutral (−588.45), isolation with bidirectional migration and population growth (−803.30), and isolation with population growth and no migration (−2026.93) models. The final model tested, split-bidirectional-migration, had an intermediate likelihood of −286.49. The split-with-no-migration model was unstable under all conditions tried, and we could not get it to run to convergence. We provide jackknifed estimates and CIs for the best-fitting, split-with-migration model in Table 1.

Table 1 Demographic model parameters from the δaδi split-migration model (variables in first column) and estimates in biological units (defined in final column), with 95% CIs determined by jackknifed datasets. The two migration rates use the two different effective population sizes in their calculation.

Model parameters	Parameter (+95% CI)	Estimates (+95% CI)	Lower–upper bounds	Biological units	
nu1 (pop size McKay’s)	3.52 (±0.54)	109,330 (±16,790)	92,540–126,120	Individuals McKay’s	
nu2 (pop size snow)	5.95 (±1.79)	184,991 (±55,523)	129,467–240,514	Individuals snow	
T (split time)	1.44 (±0.37)	241,491 (±62,429)	179,061–303,920	Years	
m1 (migration)	1.65 (±0.39)	2.90 (±0.10)	2.8–3.0	Individuals using nu1	
m2 (migration)	1.65 (±0.39)	4.90 (±0.35)	4.6–5.2	Individuals using nu2	
theta	249.97 (±32.71)a	31,072 (±4,066)a	27,006–35,138	Ancestral population individuals	
Note:

a Nref.

Discussion

Our data provided sufficient variation to answer fundamental questions about these two recently diverged taxa, despite a lack of fixed genetic differences and evidence for moderate levels of gene flow. Thus, our study adds to evidence showing the utility of UCEs for illuminating key evolutionary attributes among populations with shallow levels of divergence (Table 2). These data also provide a direct comparison to markers previously used to investigate recent divergence (i.e., mtDNA, AFLPs) for these same taxa (see below). As UCEs are used more frequently for population genomics, in addition to systematics, new actions become desirable (Table 2). Some of the key approaches are: sequencing at increased depth, genotyping individuals (determining both alleles of a locus), implementing GQ filters, accounting for recombination, improving mutation rate estimates, and implementing population genomics analytical pipelines rather than those oriented more typically toward systematics. Questions often differ at population levels, but researchers are successfully applying a variety of approaches that demonstrate the utility of UCEs in population genomics (Table 2).

Table 2 Some bioinformatic and analytical attributes typical of population genomics studies and some of the variation among researchers in applying them to different questions using UCE data.

Population genomics attribute	Smith et al. (2014)	Harvey et al. (2016)	Zarza et al. (2016)	Oswald et al. (2016)	This study	
Genotyping	No	Yes	Yes	Yes	Yes	
GQ filters	No	Yes	Yes	?	Yes	
Recombination	No	No	No	No	Yes	
Substitution rates	Yes	Yes	No	Yes	Yes	
Population differentiation	Yes	Yes	Yes	Yes	Yes	
Gene flow rates	Yes	No	No	Yes	Yes	
Effective population sizes	Yes	Yes	No	Yes	Yes	
Heterozygosity	No	Yes	No	No	Yes	

In considering UCEs as a class of markers that subsamples the genome, it is useful to note that our estimated substitution rates (mean of 6.75 × 10−10 substitutions per site per year) are roughly an order of magnitude slower than the mutation rate estimated across the entire genome of three generations of Ficedula flycatchers (Smeds, Qvarnström & Ellegren, 2016). This is perhaps not surprising given the conserved nature of these loci. Using a very different method to estimate substitution rates (scaling UCE results to an mtDNA molecular clock), Harvey et al. (2017) estimated rates of 1.74 × 10−12 to 2.32 × 10−11 substitutions per site per year for UCE loci, one to two orders of magnitude slower than comparable RAD-seq data from the same animals and also slower than the rates we estimated here for buntings. In addition to differences in methodology, Harvey et al. (2017) had shorter loci on average (mean locus length 604 bp) than loci in our study. Nevertheless, more study of substitution rates in loci with UCEs is warranted because these estimates are important when converting modeled demographic parameters into biological units. The effect of our estimated substitution rates on our demographic estimates (if, e.g., our substitution rate estimates are wrong) is that for some, they are positively correlated; lower substitution rates would drive effective population sizes and split times lower (Ne: nu1, nu2, and Nref and T in Table 1). Migration rate (m) estimates in Table 1 are unaffected by substitution rates.

The nucleotide diversity levels that we observed are approximately an order of magnitude lower than typical levels across the avian genome (0.0011–0.005; Ellegren, 2013). This is likely the result of purifying selection acting on UCE loci, effecting an apparent lower substitution rate. Our values are more similar to the values for Z-linked loci in other bird species (Balakrishnan & Edwards, 2009; Huynh, Maney & Thomas, 2010; Lavretsky et al., 2016).

When applied to this relatively recently derived pair of taxa, UCE results raise the question of whether McKay’s bunting is a full biological species. Although McKay’s bunting is taxonomically recognized as a species, this dataset shows substantial levels of gene flow (see Wright, 1943; Cabe & Alstad, 1994; Winker, 2010), and the lack of fixed alleles is surprising given that we sampled thousands of loci and four individuals from each of two putative species; we would expect several fixed differences to occur by chance through neutral processes. There are some noted plumage differences between the two taxa (Maley & Winker, 2007). But while our results enabled 100% diagnosability (which might decline with broader sampling; it was 96.5% using AFLPs in Maley & Winker, 2010), they also suggest widespread genomic similarity between McKay’s and snow buntings (e.g., relatively low FST). Given phenotypic differences between the taxa, it seems likely that there are fixed allelic differences in portions of the genome not included in our data that could be detected by more extensive surveys of each species’ genome. The status of the taxa as biological species, however, is more likely to hinge on gene flow (i.e., the geographic partitioning of traits that may be responding to adaptation is not equivalent to speciation).

There are reports of male McKay’s buntings present outside their breeding range and possible hybridization between McKay’s and snow buntings (Sealy, 1967, 1969). Snow buntings are also common on the breeding range of McKay’s buntings at St. Matthew Island prior to and during early portions of the breeding season, although most individuals leave before fledging (Winker et al., 2002). Just one pair of snow buntings has been recorded on the island during fledging (Winker et al., 2002). Observations thus suggest the possibility of hybridization; our data provide a confirmation and a quantification of it. The levels of gene flow that we found, 2.8–5.2 individuals per generation (Table 1), seem rather high for two putative biological species (Rice & Hostert, 1993; Hostert, 1997; Winker, 2010). Further study will be needed to determine species limits between these taxa, including larger sample sizes, broader genomic coverage, and proper caution for interpreting genomic results in terms of species delimitation (Robinson et al., 2014; Sukumaran & Knowles, 2017).

In comparing UCE-based estimates of demographic parameters with those based on mtDNA sequence (Maley & Winker, 2010), we find little overlap (Table 3), and our UCE-based split-time estimate is an order of magnitude earlier. Although effective population size estimates for McKay’s buntings are close (though non-overlapping), those for snow buntings are one-to-two orders of magnitude smaller, a difference that is only partially explained by differences in effective population size for autosomal and mtDNA estimates. These differences may also be driven by the different selection regimes operating on the two marker classes. For example, purifying selection on UCEs will result in background selection on linked variation in flanking regions, reducing (through hitchhiking) the effective population size (Charlesworth & Charlesworth, 2016). Previously, mtDNA results suggested that gene flow was highly asymmetric (Maley & Winker, 2010), concordant with what was likely a post-glacial introgression of McKay’s buntings into snow buntings during a snow bunting range expansion into Beringia. Our UCE-based estimates have much narrower confidence limits (and without the strong asymmetry found in mtDNA; Table 3), but they do suggest moderate levels of gene flow between the two species.

Table 3 Comparing bunting UCE results of demographic estimates with those obtained using mtDNA by Maley & Winker (2010).

Parameter	UCEs	mtDNA	
Split time	179–304 Kyr	18–74 Kyr	
Ne McKay’s	93–126 K	170–680 K	
Ne snow	0.13–0.24 million	6–24 million	
m	2.8–5.2	0.05–753	

Conclusions

Although more work is needed to understand demographic estimates made using UCEs relative to those obtained using other markers, UCEs provide rich, high-quality data for population genomic studies (Table 4). They are thus an important new class of genomic marker that should provide broad comparative value among diverse population genomics studies, with ever-increasing value as additional studies using UCEs (or whole genomes from which UCEs can be obtained) are conducted.

Table 4 Comparing avian UCE population genomic characteristics.

Species	# Loci	% Polymorphic	Nucleotide diversity	Heterozygosity	Source	
Tringa semipalmata	4,635	94	n.a.	n.a.	Oswald et al. (2016)	
Cymbilaimus lineatus	776	53	0.0019	n.a.	Smith et al. (2014)	
Xenops minutus	1,368	73	0.0019	n.a.	Smith et al. (2014)	
Schiffornis turdina	851	77	0.0003	n.a.	Smith et al. (2014)	
Querula purpurata	1,516	58	0.0013	n.a.	Smith et al. (2014)	
Microcerculus marginatus	1,077	60	0.0015	n.a.	Smith et al. (2014)	
Plectrophenax spp. (2)	3,431	77	0.0005	0.20–0.22	This study	
Average of 40 Amazonian species	2,416	Varied	0.0011	∼0.44 (1 sp.)	Harvey et al. (2017)	

Supplemental Information

Supplemental Information 1 Supplemental Information.

Table S1, Figures S1–S4, loci with high Fst.

Click here for additional data file.

Oralee Nudson provided invaluable guidance for supercomputer use. Matthew Miller, Mike Harvey, and Naoki Takebayashi provided helpful suggestions in bioinformatics, and Jessica McLaughlin provided a script for Fig. S2. Kevin Hawkins, Phil Lavretsky, Jeff Peters, and Ryan Gutenkunst provided help in running δaδi. We also thank Iris Cato, Maryanne Evans, Kathryn Everson, Angela Gastaldi, Jessica McLaughlin, Kendall Mills, Katie Shink, Ana Silva, Naoki Takebayashi, Sara Wilbur, and an anonymous reviewer for helpful comments.

Additional Information and Declarations

Competing Interests

Author Contributions

Data Availability

Brant C. Faircloth is an Academic Editor for PeerJ. The authors declare that they have no other competing interests.

Kevin Winker conceived and designed the experiments, analyzed the data, contributed reagents/materials/analysis tools, prepared figures and/or tables, authored or reviewed drafts of the paper, approved the final draft.

Travis C. Glenn conceived and designed the experiments, performed the experiments, analyzed the data, contributed reagents/materials/analysis tools, authored or reviewed drafts of the paper, approved the final draft.

Brant C. Faircloth conceived and designed the experiments, analyzed the data, contributed reagents/materials/analysis tools, authored or reviewed drafts of the paper, approved the final draft.

The following information was supplied regarding data availability:

The data analyzed and unique scripts are at Winker, Kevin; Glenn, Travis; Faircloth, Brant (2018): Plectrophenax bunting ultraconserved elements (UCEs) study. figshare. Fileset. https://doi.org/10.6084/m9.figshare.6453125.v1.

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
