# Peer review of "Ultraconserved elements (UCEs) illuminate the population genomics of a recent, high-latitude avian speciation event"

_PeerJ, doi:10.7717/peerj.5735_

## Round 0.1 · original submission · Major Revisions

Dear authors

Please find the comments of our reviewers. For revision, please implement the recommendations in your revision, as we will send the ms to the original reviewers.

Try to avoid too much technical language, or explain your acronyms, so that a reader can understand the ms.

Regards
Michael Wink
Academic editor

Reviewer 1 ·

Basic reporting

The writing here is clear and concise but most of the jargon is above my pay grade. I have no doubt that experts in genomics analyses will understand all this but it’s a daunting task for the non-expert to understand the Results section. The Discussion, however, makes it very clear what those results can and cannot tell us.

Experimental design

Seems fine and highly appropriate but someone more familiar with the methods should evaluate this

Validity of the findings

The findings seem entirely valid given the results but I do not have enough knowledge to evaluate the validity of those results

Additional comments

This looks to me like an interesting and very valuable study not only for showing the value of UCEs but also helping to determine the extent of genetic difference between McKay’s and Snow buntings. As the authors say, more work will be needed to determine where the consistent genetic difference are between these two species, especially with respect to identifying the genes responsible for the plumage difference. A few recent studies of other sister taxa of birds have been able to do this. This main value of this study seems to be in the evaluation of UCEs as a population genetics tool.
Also:
18-34 I feel there is too much jargon in this abstract. As it stands it will only be comprehensible to experts in this sort of genomic analysis. I suggest rewording or defining: orthologous, ultraconserved elements, marker sets, analysis pipelines, 30x sequencing coverage, significant genetic structure
51 this is unnecessary hyperbole. Powerful computers are not in any way strained by these datasets, they just take more time to analyze
Table 1. I would like to see more details in the figure caption as I do not understand the difference between parameters and estimates. It’s also not clear why you present lower/upper bounds and 95%CIs because they tell the same story. 95%CIs

·

Basic reporting

This is an overall well wrote and clear manuscript but a few paragraphs could be improved for readability (e.g. lines 83-87, 158-161, 169-171).

The structure of the manuscript conforms to the journal standards. Although the introduction is very relevant in the context of UCEs, the manuscript could be improved with more detail about the target species, including the known range distribution with the locations of the used samples.

The figures seem fine although they could just be considered as supplementary material, table 3 and 4, could be included as supplementary material and not in the main text. It was supplied raw data and code of the main demographic analysis, but I would advise the authors to add the scripts used for bioinformatic analyses and all the analyses code into the figshare link they provided for the raw data, so there would be no need to include the code as supplementary material. Additionally, I advise the authors to add a figure describing the six analyzed models (see below).

Although this manuscript is showing the advantage of using population genetic methods with UCE data, the results or absence of these don't necessarily support that well that UCE data should be used directly for population genetics. Below I comment on this and provide a few suggestions on how still UCE markers can be very useful.

Experimental design

This is original research within the scope of the journal, and although it is a short paper, it has important relevance for the studied species and for the understanding of UCE limitations with population genetics analyses.

The description of the methods was very thorough from bioinformatics to the demographic analyses, which helps anyone that would like to perform similar studies. However, the description of the models was less clear and that could be improved by drawing the scheme for which model (e.g. Robinson et al. 2014, BMC Evolutionary Biology).

This figure could help simplify the detail describing these methods since this section is confusing. This is just a suggestion but the authors could refer to the simpler model as no isolation, the simple split model as strict isolation and the models with migration and growth as isolation with migration and population change, with the sixth model as isolation with bidirectional migration. Additionally, I suggest to the authors to add, even if as supplementary, a figure of the spectrum which can show how informative the folded SFS is.

The optimization procedure could be more explicit in describing how many repeats were performed when varying the parameters within bounds, and please explain why three runs with the highest observed likelihood value would be enough. The same issue for jackknifing with only 10 replicates since this seems insufficient, but not sure if the authors did more and realized the difference between running 10 or 50 times yielded similar results. Also, not sure if averaging the highest five values when getting variable likelihoods is standard practice, but if so please add a reference (line 190). For better readability, the results for model selection could also be described in a table with the model and its respective maximum log composite likelihood values.

In relation to the sequence data analyses that were not shown, I suggest the authors to either remove the recombination analysis, because it doesn’t correspond to the main findings of the paper, or to include but with a plot that shows the lack of convergence by IMa2p.

Validity of the findings

Considering the difficulty in dadi with the “split-with-no-migration model” (line 264) and the no convergence with the sequence based analyses IMa2p, I would personally, feel more comfortable in acknowledging the presence of high gene flow between these species if there was another method corroborating these results, like fastsimcoal2 which is also based on SFS. Regardless, I see great value in this work as a possible example of a study suggesting evidence that described species with a few morphological differences should possibly be considered more as highly differentiated populations than as species.
I commend the authors for being cautious in declaring the need for more work to properly infer if these should be considered as different species, but suggest the authors to mention in the discussion how this could be done. This could possibly be done using the same UCE dataset but including a few more closely related species, like a species delimitation study with Calcariidae species including validation methods like BPP, or methods that use Bayes factors to properly test different hypotheses for the number of species (e.g. *BFD SNAPP and BFD StarbBEast2). I suggest this because the big advantage of sequence capture methods, like UCEs, is the flexibility of analyzing phylogenomic data and population genetics data, so a future study that uses the same markers but including samples from different species would exemplify this advantage of UCEs.
However, the authors could discuss a bit more the limitations in applying UCEs to population genetic studies, since using one lane to sequence 8 individuals from 2 species and only be able to perform SNP-based analysis (without including an outgroup), would probably make the study not the most cost-efficient, unless the same data is being used to address other questions. For the number of samples used and given the use of only SNP data, probably RadSeq data would have been better, but I can see that the goal was to also explore demographic analyses with haplotype data, which it seems these did not converge. If more individuals had been sequenced, another possible sequence-based approach would be using Extended Bayesian skyline plots (Heled and Drummond 2008).
Given this, I suggest the authors to reconsider what they think is the main take-home message in the conclusions, personally I think it is really interesting the evidence that these two described species might have more gene flow than expected to be considered as different species implying the need of a proper species delimitation study with integrative taxonomy.

Additional comments

Some additional comments:
• Not clear why it should be removed the FASTA sequences from non-UCE loci (lines 118-119), these sequences could provide more informative loci if they would still align across species.
• For consistency, please add the version of used software where this is not specified.
• I assume the description of subselecting one SNP per locus is repeated in line 146 and 166, and that this step was not done before and after converting the SNPs into a biallelic format if that is the case I suggest removing line 146 and edit the sentence.
• Although it is mentioned in the results, I would add to the methods that a DAPC was done, and possibly complement it with a PCoA, even if it is just to refer these plots to the supplementary material, so it can be visualized the 100% diagnosable samples to clusters.
• As suggested, rewrite the description of the models since it is not clear the difference between 4 and 5, which I assume is that one includes population change parameters. The expression "five basic and one derivative model" seems unusual to me, these models are more or less complex in number of parameters even if the 5th model was not in the example functions in dadi. For better comprehension of the alternative models, it could be better to enumerate them by complexity in parameters.
• I acclaim the authors for explaining how they obtained each value to convert estimated parameters to time and population size, but given the filtering performed during bioinformatic processing, I think it would be helpful to mention what is the total sequence size used to convert the substitution rate. In line 290-291, it is mentioned the difference in loci size as a comment for the difference in substitution rates, but ideally this wouldn’t affect the results when you take into account the total sequence size, so this comment seems unnecessary.
• The authors mention this is a study that adds to the utility of UCEs when looking to population data (line 271), please add a few other examples where analyses at similar scales have been done.
• Considering the reduced number of samples used and these were from very few populations, it could be possible the DAPC result is a bit biased by lack of diversity representing the distribution of the samples, so maybe there is a need for some caution about how much it can be said about the 100% diagnosability.
• The species names in figure 2 legend should have the genus name or at least "P.".

---

## Round 0.2 · Major Revisions

Dear authors

Uur reviewer still recommends changes to your ms.

Kind regards
Michael Wink
Academic Editor

·

Basic reporting

Overall it seems to me the authors have improved and clarified a lot of the issues that I previously mentioned in the first review. However, I still have some minor points that I think need clarification.

Experimental design

The methods section about DAPC should be a bit more detailed. This is mostly a problem because of the 100% diagnosability point that is used as an argument for supporting of these species. In the DAPC tutorial (https://github.com/thibautjombart/adegenet/blob/master/tutorials/tutorial-dapc.pdf), there's a section entitled 'On the stability of group membership probabilities' that explains how 'if too many PCs are retained this will have a destabilising effect on the coefficients estimated, leading to problems of overfit. As a result, membership probabilities can become drastically inflated for the best-fitting cluster, resulting in apparent perfect discrimination.' This is also why I suggested to additionally provide a PCA/PCoA, since discriminant functions are linear combinations of variables which optimize the separation of individuals into pre-defined groups, so the potential "lack" of clustering that a PCoA would show, might be more realistic than the inflated clustering the DAPC might be suggesting, if the retained number of PCs was much higher than the number of analysed samples. If this was something that it was taken into consideration, at least I request the authors to add the number of retained PCs.

There's another issue that I should have realized in the first review. In table 1, there are two migration parameters when the best model is a split with one migration rate between populations, for which the authors get the number of migrants by multiplying by the population size of each taxon. However, the migration rate estimated from this model can only provide the number of migrants between populations without an asymmetric gene flow. This should actually be calculated in relation to Nref and not to the estimated with the population size of each taxon, I found this topic in dadi's forum to be useful: https://groups.google.com/forum/#!topic/dadi-user/AY_4NoEjsx8. Additionally, I urge the authors to include the information of how many sites were considered to make these calculations, given the concern to think about what would be the best substitution rate and generation time, this is the only information that is not explicitly stated for this purpose.
Since the authors prefer to keep the section about recombination within loci, I suggest this section be moved in the methods to be before the demographic analyses, following the order as it is in the results.

Validity of the findings

The issues still pending about the validity of the findings are related with my two comments in the previous section. Since the number of migrants per generation should be corrected, I think it's likely the number of migrants per generation would be two orders of magnitude lower, which will change the discussion of the findings. This suggests that maybe gene flow between these taxa is not as high as the 2-3 migrants, which tend to be more common between well-connected populations than between supposedly well-differentiated species.
Since I am also not so certain about the 100% diagnosability of DAPC without knowing if the sampling size was considered (or without observing these data with a PCoA or a structure plot), there might be more to comment in the discussion about evidence of why these population genetics results might support or not the validity of these species. Given the relatively recent split time and the evidence for gene flow (although now maybe low), I think the difficulty to do proper species delimitation within this group, as referred by the authors in response to my comments, could also be mentioned in the discussion. I personally think it is also the lack of an outgroup, even to get an unfolded SFS which could be more informative, that is limiting better conclusions from the population genetics side of analyses that can be executed when trying to do proper species delimitation.
This is just another suggestion that I should have provided initially, but since the authors estimated nucleotide diversity using a concatenated alignment in MEGA, and the fact these species seem to be in the "grey zone of speciation" (sensu Reux et al. 2016), maybe it would also be worth it to provide dxy and da values between species beside the significant but really low Fst value.

Additional comments

These are just some minor issues that could be improved:
- The last sentence of the introduction reads much better but some commas would improve readability.
- The structure of subsections like 'Laboratory.—' in lines 96, 115, 148 seems to be different from most papers I have seen from peerJ.
- In line 113, I personally find it unnecessary to abbreviate 150 bp pair-end sequencing since it just adds another abbreviation that should be clarified.
- There were a few references in the manuscript for just Supplemental Information without referring to the number of table/figure (e.g. line 188, 249).
- In line 215, it is not clear what is (hit table, csv) referring to.

I apologize to the authors for not identifying these two main issues I detected by reviewing this manuscript a second time. I think this a relevant paper that will be a stepping stone for future research done with UCEs and demographic analyses, and that's why I think these issues should be clear and well done for others to follow this work.

---

## Round 0.3 · accepted · Accept

Dear authors

Congratulations. Your revisions are adequate and your ms is accepted. Thanks for submitting your interesting work to our journal.

Kind regards

Michael Wink
Academic editor

·

Basic reporting

No comment

Experimental design

No comment

Validity of the findings

No comment

Additional comments

Thank you for the thorough responses and apologies for my own confusion in calculating the number of migrants from your estimates, I seem to have confused how to use mij and Mij to get the number of migrants.